# Genomic Medicine in Canine Periodontal Disease: A Systematic Review

**DOI:** 10.3390/ani13152463

**Published:** 2023-07-30

**Authors:** Carolina Silva, João Requicha, Isabel Dias, Estela Bastos, Carlos Viegas

**Affiliations:** 1Department of Veterinary Sciences, School of Agricultural and Veterinary Sciences (ECAV), University of Trás-os-Montes e Alto Douro (UTAD), Quinta de Prados, 5000-801 Vila Real, Portugal; carolinasilva95@hotmail.com (C.S.); jfrequicha@utad.pt (J.R.); idias@utad.pt (I.D.); 2CECAV—Centre for Animal Sciences and Veterinary Studies, University of Trás-os-Montes e Alto Douro (UTAD), 5000-801 Vila Real, Portugal; 3AL4AnimalS—Associate Laboratory for Animal and Veterinary Sciences, 1300-477 Lisboa, Portugal; 4CITAB—Center for the Research and Technology of Agro-Environmental and Biological Sciences, University of Trás-os-Montes e Alto Douro (UTAD), 5000-801 Vila Real, Portugal; ebastos@utad.pt; 5Inov4Agro-Institute for Innovation, Capacity Building and Sustainability of Agri-Food Production, 5000-801 Vila Real, Portugal; 6Department of Genetics and Biotechnology, School of Life and Environmental Sciences, University of Trás-os-Montes e Alto Douro (UTAD), 5000-801 Vila Real, Portugal

**Keywords:** dog, genomics, clinical, periodontal disease, animal model, systematic review

## Abstract

**Simple Summary:**

Periodontal disease is one of many conditions that affect the oral cavity of companion animals and humans. Through animal models, namely the dog, it has been possible to increase our knowledge about this disease. Currently, one of the most explored areas is genomic medicine. Through it, it will be possible to implement individualized and early strategies to prevent the development of periodontal disease. In light of the limited existing information, the aim of the present study was to systematically review the existing scientific literature regarding genomic medicine in canine periodontal disease, focusing on the genes already studied and their probable potential. Six articles were selected and analyzed in detail. Only in two of them was it possible to determine that the studied genetic variations could be potential biomarkers to determine the susceptibility to the development of periodontal disease. This fact reinforces the need for further studies in this field, since it is clear that the future of human and veterinary medicine will involve an approach based on the genetic characteristics of each individual.

**Abstract:**

Genomic medicine has become a growing reality; however, it is still taking its first steps in veterinary medicine. Through this approach, it will be possible to trace the genetic profile of a given individual and thus know their susceptibility to certain diseases, namely periodontal disease. This condition is one of the most frequently diagnosed in companion animal clinics, especially in dogs. Due to the limited existing information and the lack of comprehensive studies, the objective of the present study was to systematically review the existing scientific literature regarding genomic medicine in canine periodontal disease and determine which genes have already been studied and their probable potential. This study followed the recommendations of the PRISMA 2020 methodology. Canine periodontal disease allied to genomic medicine were the subjects of this systematic review. Only six articles met all of the inclusion criteria, and these were analyzed in detail. These studies described genetic variations in the following genes: interleukin-6, interleukin-10, interleukin-1, lactotransferrin, toll-like receptor 9, and receptor activator of nuclear factor-kappa B. Only in two of them, namely interleukin-1 and toll-like receptor 9 genes, may the identified genetic variations explain the susceptibility that certain individuals have to the development of periodontal disease. It is necessary to expand the studies on the existing polymorphic variations in genes and their relationship with the development of periodontal disease. Only then will it be possible to fully understand the biological mechanisms that are involved in this disease and that determine the susceptibility to its development.

## 1. Introduction

Periodontal disease (PD) is a set of dental inflammatory diseases initiated by oral microbiota on the tooth surface [1]. PD is staged based on the clinical observations of gingivitis and periodontitis [2]. It is a progressive disease, initially presenting as gingivitis and progressing to periodontitis [3,4,5]. Gingivitis is characterized by reversible inflammation and gingival redness without the loss of connective tissue attachment or alveolar bone. If this condition is left untreated, most, but not all, cases of gingivitis progress to periodontitis. Periodontitis is a severe chronic inflammation of the supporting tooth tissues that causes the loss of connective tissue attachment and alveolar bone, possibly resulting in gingival recession, oronasal fistula, periapical lesions, tooth mobility, and tooth loss [6]. 

This is one of the most frequently diagnosed diseases in companion animal clinics [4,7], which represents a problem with a great health and economic impact. According to some authors, around two years of age, about 70% of cats and 80% of dogs already have PD [6]. In dogs, the most frequently affected breeds are the small and miniature breeds [8], and PD is more frequent as age advances [9]. In addition to companion animals, PD is also one of the most common diseases of the oral cavity in humans [2,7,10,11]. The impact of PD in human and veterinary medicine is widely recognized, with serious implications for systemic health [12].

The etiology of this disease is multifactorial, as behavioral, microbiologic, environmental, systemic, and genetic factors contribute to the susceptibility and clinical expression [13]. Recent research has shown that genes play a key role in the predisposition and progression of PD [14], however, the vast majority of this research is in human medicine and the results are relatively inconsistent [15,16].

The dog is widely used as a model to study periodontal disease as it has characteristics that make it an ideal model [2]. However, to date, the role that genetics plays in the development of PD in the dog is still unknown [17,18], and there are still very few studies that have analyzed this possible relationship. Even so, it should be noted that, as in humans, genetic variability may be one of the factors that justify the susceptibility to disease of some individuals compared to others [19,20].

Genome wide association studies (GWAS) are conducted on hundreds of thousands of genetic variants across many genomes to find those statistically associated with a specific trait or disease [21]. Genome-wide canine single nucleotide polymorphism arrays have been developed and increasing use of these arrays to map disease loci in dogs have been noticed. GWAS have been performed for different diseases in the dog, which are described below.

GWAS have been performed in different diseases of the dog including canine systemic lupus erythematous related disease complex [22], heritable osteosarcoma [23], and squamous cell carcinoma of the digit [24]. Further canine work has centered on the importance of using a reference panel with 365 genome sequences to improve the power of GWAS [25] and performing meta-analysis after breed specific GWAS [26].

Despite the above-mentioned studies, the association between canine PD and genomic medicine has not yet been widely investigated. Thus, the exclusive focus of this work was in this association. In fact, to our knowledge, this is the first systematic review in genomic medicine of canine PD. The aim of the present study was to conduct a review of all the articles existing on this topic to better understand the central role that genetics plays in the development of PD. Given the limited number of studies regarding genomic medicine in canine PD, this review will certainly contribute to the understanding of which genetic factors have already been analyzed, and consequently promote further research on other potential biomarkers.

## 2. Materials and Methods

The PRISMA 2020 (Preferred Reporting Items for Systematic Reviews and Meta-Analyses) guidelines [27] were used to conduct the writing of this paper. Thus, it was possible to ensure that the writing of this review was conducted systematically and without bias. The study has not been registered in PROSPERO since it is not for human health.

### 2.1. Inclusion and Exclusion Criteria

The inclusion criteria for this study were articles written in English, published between 2002 and 2022 in indexed journals with relevant information about the genomic component in canine PD.

The studies that (1) were not written in English, (2) did not describe a clear relationship between genomic medicine and canine PD, (3) were repeated, (4) were about a species other than dog (*Canis lupus familiaris*), (5) were about microbiology and (6) were textbook chapters, technical manuals, conference proceedings, and abstracts were excluded.

Considering these criteria, we read the selected studies in full and only considered those that contained explicit information.

### 2.2. Sources of Information and Search Strategy

We searched for studies in the electronic databases Science Direct/Elsevier and PubMed (National Library of Medicine, National Institutes of Health, Bethesda, MD, USA), between 1 January 2002 and 31 December 2022. The terms used for the search were “genomics OR genetics”, “canine OR dog”, and “periodontal disease”.

### 2.3. Risk of Bias in Individual Studies

Assessment of the risk of bias (scientific quality) of each article included was performed in accordance with the “Newcastle–Ottawa tool to assess risk of bias”. The included case–control studies were specifically analyzed with the Newcastle–Ottawa Quality Assessment Form for Case–Control Studies. This assessment was prepared independently by two researchers.

### 2.4. Data Analysis

A descriptive analysis was carried out on the important information and results found in the articles selected for the preparation of this review.

## 3. Results

After the entire search process, 720 studies were found on genomics in canine PD. Of these 720 studies, 459 (63.75%) of them were obtained from the Science Direct/Elsevier and 261 (36.25%) from PubMed. Duplicates were removed and the abstracts were re-evaluated. By evaluating only the title and respective abstract, we were able to exclude 494 studies. The exclusion of these studies was due to non-compliance with the defined criteria. Most of them were excluded because the species under study was different from the dog and had an approach focused on medical microbiology rather than genomics medicine. After reviewing the 31 studies, only six still met all of the initially defined criteria (Figure 1).

We considered the development of scientific articles focusing exclusively on canine PD genomics medicine as an essential aspect in the evaluation of these studies. All studies selected here were considered as case–control. Given the limited number of studies found and the differentiating research element between each of them, the authors opted for an individual approach in this systematic review. Additionally, this review demonstrated some heterogeneity between studies, as evidenced by the elements under analysis or the final conclusions. The genetic variants analyzed came from the interleukin-6 (*IL6*) gene, lactotransferrin gene (*LTF*), interleukin-10 (*IL10*) gene, interleukin-1 (*IL1*) gene, toll-like receptor 9 (*TLR9*) gene, and receptor activator of nuclear factor-kappa B (*RANK*) gene. In all of the analyzed studies, the candidate gene approach was followed. This is an essential instrument to find the genetic risk factors for complex diseases like PD [28]. Table 1 summarizes the main characteristics and the main findings of each of the articles that were selected and analyzed (in chronological order).

### 3.1. IL6 Gene Variants

One of the studies included in this systematic review performed a molecular analysis of the *IL6* gene in order to identify genetic variations and verify its association with PD [12]. As a case–control study, the population was separated into two groups. The control group included 45 dogs with healthy periodontium and the cases group included 25 dogs with a PD degree range of gingivitis to advanced periodontitis. Two regions of the *IL6* gene were amplified by PCR, and all of the amplified fragments were sequenced and genotyped, resulting in three new genetic variations being identified. The three new genetic variations were *I/2_g.564T>C* located in exon 2, *I/5_g.105G>A*, and *I/5_g.440G>A* located in exon 5. No statistically significant differences were detected between the control and PD cases group.

### 3.2. LTF Gene Variants

The aim of this study was to molecularly analyze the dog *LTF* gene to identify possible genetic variations and verify the association with PD [29]. The population in study was separated in two groups: the case group included 40 dogs with PD and the control group included 50 dogs with healthy periodontium. The authors performed PCR amplification of three regions of the *LTF* gene. All amplified fragments were sequenced and genotyped, allowing for the identification of eight new single nucleotide variations. In the first fragment, two variations located in intron 2 were identified, but the low incidence of these variations was not statistical relevant. The animals genotyped in this study did not present any variations localized in the second region analyzed. The other six variations were detected in the third fragment. For the variations included in the case–control study, the distributions of genotypic and allelic frequencies were analyzed. No statistically significant differences were found between the control group and the group of cases with PD, and no association between the genetic variations and the susceptibility to PD was established.

### 3.3. IL10 Gene Variants

In [30], the authors delineated hypothesizing that the dog *IL10* gene may present variation, mainly in the 5′-flanking region, that can influence the *IL10* regulatory role, and consequently, the PD susceptibility. After the oral exam, the dogs selected were distributed in two groups. The case group was composed of 40 dogs with PD and the control group was composed of 50 dogs with healthy periodontium. After the sample collection and DNA isolation, two fragments were defined in the *IL10* gene region described above. Both of them were amplified by PCR and sequenced, originating the identification of six new nucleotide variations and 3-nucleotide deletion. No statistically significant differences were detected in the genotype and the allele frequencies of the seven new genetic variations between the PD cases and control groups. However, it was possible to find that one of the variations led to an amino acid change in the putative signal peptide. This amino acid change may influence the *IL10* protein functionality.

### 3.4. IL1A and IL1B Genes Variants

One of the case–control studies [31] aimed to molecularly analyze the dog *IL1A* and *IL1B* genes in order to identify possible variations that could be related to the development of PD. With this aim, they selected 40 dogs with periodontal disease and 50 healthy dogs. Blood samples were collected from each dog and their DNA was isolated. Subsequently, two different fragments of the *IL1A* gene and only one of the *IL1B* gene were studied. The amplified fragments of the two genes were sequenced and genotyped. Eight genetic variations were identified, seven of them in the *IL1A* gene and one in the *IL1B* gene. The differences considered statistically significant found between the two groups under study led to the conclusion that *IL1* genetic variants may be important in PD susceptibility in canines.

### 3.5. TLR9 Gene Variants

One study [32] investigated the possible role that sequence variants in *TLR* genes could play in the predisposition and progression of PD, building on previous studies in human medicine. This case–control study included 90 dogs (50 control and 40 cases) from which their DNA was obtained. After this, a fragment of the *TLR9* gene was amplified by PCR and sequenced. The obtained results showed that two of the genetic polymorphisms analyzed in the *TLR9* gene are good candidates for determining individuals with a predisposition of developing PD.

### 3.6. RANK Gene Variants

The most recent study [33] analyzed in this systematic review aimed to determine a potential association between genetic variations in the *RANK* gene and PD. A case–control study was designed and DNA samples were collected from 90 dogs, which were divided into two study groups (50 controls and 40 cases). The obtained results showed that there were no statistically significant differences between the control group and the group with PD. However, this study allowed the authors to discover four new intronic variations that were stored in GenBank.

### 3.7. Risk of Bias Analysis

Table 2 shows the results of the risk of bias analysis for each of the six selected case–control studies [34], revealing the same classification for all of them (total score of 16 out of 20). The elements that were lacking in the considered studies were non-response rate, corrected statistics, genotyping blind to case–control status, and proposed inheritance model.

## 4. Discussion

Periodontal disease is a highly prevalent inflammatory disease that occurs in periodontal tissues as a result of oral bacterial film [30]. Both dogs and humans are susceptible to this illness, which may have serious systemic health implications [2]. The systemic disorders most associated with canine PD are cognitive dysfunction [35], cardiac disease [36,37], and renal disease [38,39], which have a significant impact on quality of life.

The link between canine PD and canine cognitive dysfunction stems from the association made with PD and Alzheimer’s disease in humans, as canine cognitive dysfunction is considered a natural animal model of human Alzheimer’s disease. It is theorized that the chronic inflammatory state characteristic of oral dysbiosis in patients with periodontal disease leads to disruption of the blood–brain barrier, cytotoxin- and pathogen-induced brain damage, and the accumulation of neurotoxic β-amyloid [35].

Some epidemiological studies have suggested that PD is associated with cardiovascular diseases. The most frequently associated pathologies are endocarditis and cardiomyopathy. The most likely explanation for this association is the chronic inflammatory stage and bacterial proliferation in the oral cavity of dogs with PD, which consequently has systemic implications [36,37]. Nevertheless, further research is needed to better clarify this association and explore possible explanatory mechanisms [37].

In the specific case of renal disease, the severity of canine PD is associated with systemic inflammation and changes in serum creatinine and blood urea nitrogen concentrations [38,39]. However, further studies are needed to clarify this association.

In humans, in addition to renal [40], cardiac [41,42,43,44], lung [45], and liver [46,47,48,49,50] pathologies, PD is also associated with osteoporosis and preterm low weight in newborns due to the translocation of periodontal pathogens into the fetoplacental unit and inflammatory mediators [51], which are the main systemic implications associated with PD [52,53,54]. In the specific case of osteoporosis, the most common metabolic bone human disease, the systemic bone resorption may be closely correlated with osteoporotic change in the jaws and alveolar bone loss [55]. In addition, the existence of pro-inflammatory mediators detected in both pathologies, more specifically *IL1* and *IL6*, may influence osteoclast differentiation and activity [56]. Furthermore, these mediators may aggravate the tissue response to periodontitis and further promote the inflammatory response [57]. The association of PD with systemic health is an impetus for identifying genetic variations that put patients at risk, and therefore promotes further research.

It is not possible to talk about genetic investigations without addressing the concept of multi-omics analysis. The expression “omics” refers to a novel method of analysis that enables the investigation and subsequent attempt to comprehend a biological system as a whole while simultaneously analyzing many molecules. This new method has been proven to be helpful from etiopathogenetic, screening, diagnostic, as well as therapeutic perspectives in the study of both physiological and pathological processes [58,59,60]. The multi-omics analysis is already widely recognized in human medicine. However, it is now taking its first steps in veterinary medicine, with the aim of promoting the health of animals and people as well as increasing production in animal husbandry [61]. This approach focuses on several areas, namely metabolomics, transcriptomics, lipidomics, proteomics, and clinical genomics, allowing for an in-depth molecular understanding of the individual, depending on the study site where the samples are collected [62]. Depending on the type of study intended and the material to be analyzed, the samples often used are blood, gingival crevicular fluid, gingival biopsy, and saliva [63,64], which later gained a particular prominence. It is a biomarker with great potential due to the fact that its collection is considered as non-invasive and easily accessible [65,66]. Since it is in direct contact with the periodontium, its analysis will allow one to establish the current disease activity [66,67]. PD has been the subject of multi-omics analysis in human medicine, thanks also to its recognized impact at a systemic level, and the data obtained from this analysis are considered promising for treatment success [68]. However, no multi-omics analysis has yet been performed on canine PD, which is one of the most frequently diagnosed pathologies in companion animal veterinary medicine [4,7].

In the 1990s, the first evidence that genetics contributed to periodontal disease appeared. With this new evidence, concepts like susceptibility and predisposition to PD were introduced [69].

In humans, chronic periodontitis was estimated to have a 50% hereditability rate [70] and different systematic reviews have demonstrated the strong association between genetic factors and PD [71]. Certain genetic polymorphisms have been studied for their association with chronic periodontitis including several interleukin genes, the vitamin D receptor, the *FcγRIIIb-NA1* gene, the tumor necrosis *factor-β* gene, and several human leukocyte antigen variants [16,69,72,73,74,75]. In addition to these, Demmer et al. [76] conducted a comprehensive genomic analysis to demonstrate how the gene expression at healthy and diseased periodontal locations differed. When compared to health, they discovered thousands of genes that were either up or downregulated [76].

Cytokines have a significant role in the immunopathology of PD [77,78]. The existence of cytokine gene polymorphisms frequently affects the cytokine expression profile and may therefore play a significant role in PD resistance and vulnerability [79]. By penetrating the gingiva, bacterial cell surface chemicals and released products trigger cytokine reactions in PD cases [15]. In the early phases of gingival inflammation, macrophages are probably the most significant source of these mediators, however, other leukocytes, like the B cell, may be the main source in more advanced PD [15]. The expression of particular cytokines varies according to the stage of PD. In physiological conditions, the actions of pro-inflammatory cytokines are maintained in balance by anti-inflammatory cytokines such as interleukin-4 (*IL4*), *IL10*, interleukin-11, interleukin-13, and by natural cytokine antagonists including the *IL1* receptor antagonist, soluble tumor necrosis factor (*TNF*) receptor [80]. An inadequate anti-inflammatory cytokine response may contribute to the development and progression of inflammation in chronic periodontitis [80]. According to certain research, the progression of periodontitis is characterized by high concentrations of the pro-inflammatory cytokines, *IL1* and *TNFα*, and low concentrations of *IL10* and transforming growth factor β [81]. For the various reasons already described, the genes and pathways involved in cytokine regulation have drawn the most attention in genetic investigations, particularly in PD [78].

With the aim of further investigating PD in humans, several animal models have been used [2,82]. Animal models have been used for many years because they are easy to manipulate in pre-clinical experimental models and share genetic or phenotypic characteristics with humans [83,84]. However, the dog is the ideal model for the study of PD and its use has contributed to increased knowledge in the field of periodontology [2], particularly in histopathological studies or in the development of new therapeutic approaches [85]. Healthy and diseased histological traits of periodontium are similar in humans and dogs, and PD occurs naturally in dog and is quite similar to the human pathophysiological mechanisms. In addition, PD is highly prevalent in the canine population and the existing plaque is similar to that of humans [2]. Another advantage with regard to periodontal research is that it is possible to induce experimental periodontal defects by placing silk bindings around the teeth for a period of 4–6 months or to use surgically created lesions [82]. Consequently, conducting studies with animal models is very important, given the similarity with some diseases in humans. In addition, the genetic component is increasingly fundamental to understanding the causal pathways of many diseases, as has been seen in human PD, which consequently leads to parallels with the canine species [19]. Allied to all this, we must always be based on the concept “One Health, One Medicine, One World” [86] because only then can science evolve. In the specific case of PD, the dog is the ideal model for study. However, depending on the pathology, other animal models can also be used. As an example, Clark et al. [87] analyzed the potential mechanism involved in hereditary hyperplastic gingivitis in foxes in comparison with the genetic information known about hypertrichosis occurring in conjunction with hereditary gingival fibromatosis in humans.

Despite the importance of the dog as a study model for PD, the role of genetics in the canine PD development is still largely unknown. This fact can be proven by the limited number of studies considered for the preparation of this systematic review.

In this systematic literature review, six studies were identified and evaluated. Each of the studies looked at the role that variations in different genes, namely *IL6*, *LTF*, *IL10*, *IL1*, *TLR9*, and *RANK*, would have on the susceptibility and severity of canine PD development. In human medicine, other potential biomarkers have been analyzed, namely *IL4* [14], which opens doors to continue exploring other elements and contribute to the increasingly deeper study of the genetic component in canine PD.

As above-mentioned, PD is a chronic inflammatory disease triggered by the presence of pathogenic bacteria in the subgingival region. The inflammatory response that is generated against the spread of microorganisms and for the protection of the surrounding tissues ends up generating the opposite effect. In this case, the immune-mediated inflammatory response causes tissue damage by not distinguishing pathogens from healthy resident cells. The periodontal microenvironment is characterized by persistent infection, chronic inflammatory stage, reduced immune response, and constant cell repair. Given these characteristics, senescent cells have been identified in the periodontal tissues of humans with PD. Senescent cells are characterized by a phenotype resulting from repeated exposure to lipopolysaccharides from the membrane of Gram-negative bacteria in the oral cavity of individuals with PD. Part of the pro-inflammatory cytokines that were analyzed in this review, namely *IL1* and *IL6*, are overexpressed by senescent cells [88]. Thus, it is possible to see, once again, the importance that pro-inflammatory cytokines have in the development of knowledge about PD.

*IL6* is a pleiotropic cytokine synthesized in various cells such as neutrophils and macrophages, and it plays a role in the production of acute phase proteins, among other functions [89,90]. This cytokine is essential in the inflammatory response against infectious agents, specifically Gram-negative bacteria [91]. It has been shown that *IL6* is expressed by many cell types in periodontal lesions with an inflammatory origin [92,93]. The *IL6* gene promoter region has several single nucleotide polymorphisms that have been linked to human PD susceptibility [73,94]. Thanks to these findings in human medicine, a case–control study was conducted in the dog with the aim of increasing the knowledge regarding canine PD. Although three new genetic variations in the *IL6* gene have been identified and characterized, they do not appear to influence the susceptibility to the development of PD [12]. One of the genetic variations detected, *I/5_g.105G>A,* caused an amino acid change in the *IL6* protein. The amino acid change from arginine to glutamine could potentially damage the *IL6* protein [12]. Further research is important to understand how the described genetic variations interact and their influence on protein structure and function [12].

*LTF* is an iron-binding glycoprotein included in the transferrin family [95] and is secreted in saliva, being considered the main iron-binding protein [96]. Its antibacterial effects are due to its ability to inactivate bacterial activity and capture iron [97,98,99]. Thanks to its ability to capture iron from the oral cavity, *LTF* deprives oral bacteria of an element essential for their growth and survival [100,101,102], so it appears to interact with periodontopathic bacteria [96,103]. Some studies have revealed an increased concentration of *LTF* in the gingival crevicular fluid in PD cases and an overexpression of the *LTF* gene in people with a predisposition to PD [104,105]. However, in the study performed in dogs, no evidence was found between variations in the *LTF* gene and the development of PD [29]. Nevertheless, one of the genetic variations located in exon 15, *L/15_g.411C>T*, led to the substitution of proline with leucine, resulting in an amino acid change that could potentially affect the *LTF* protein [29]. Deepening the knowledge about these genetic variations is extremely important to understand the various mechanisms and interactions of *LTF*.

*IL10* is crucial to maintain periodontal stability and health while also acting as a preventative measure against the spread of PD as it inhibits the synthesis of pro-inflammatory cytokines [80,106]. According to some authors, a decline in pro-inflammatory cytokines and an increase in *IL10* level were linked to improved clinical parameters in periodontitis [106]. A number of polymorphisms of the human *IL10* gene, namely *–1082 (–1087) A>G (rs1800896)*, *–819 (–824) C>T (rs1800871)*, and *–592 (–597) C>A* (*rs1800872*), are associated with periodontitis risk, although there are some inconsistent results [107]. Due to the strong sequence homology and similar cellular expression between canine *IL10* and human *IL10* [108,109], several studies have been conducted [110] such as the one reviewed here. However, in this case, it was not possible to establish an association between *IL10* genetic variations and canine PD [30]. The *IL10/2_g.285G>A* variation resulted in an amino acid change, glycine to arginine, in the region of exon 1 that encodes the *IL-10* signal peptide [30]. This amino acid change can have effects in immunoregulation by modifying the efficiency of secretion or the maturation process of the protein [111].

Being implicated in the pathophysiology of numerous infectious, inflammatory, and autoimmune illnesses, the *IL1* cytokine family plays crucial roles in acute and chronic inflammation [112,113]. In 1997, Korman et al. [114] analyzed the correlation between the *IL1* genotype and the severity of periodontitis. However, the association between *IL1* genetic variants and human PD has been widely studied, with more than 125 studies on this relationship [113]. Although the results are variable, the recent systematic reviews and meta-analyses have demonstrated the strong association between *IL1A* and *IL1B* with PD [113,115]. Because of these results, Albuquerque et al. [31] performed a molecular analysis of dog *IL1A* and *IL1B* genes to identify genetic variations and evaluate its possible association with PD. Similar to studies in human medicine, the work of Albuquerque et al. [31] demonstrated that the existence of *IL1* genetic variants may be important in susceptibility to canine PD. Furthermore, the *IL1A/2_g.515G>T* variation, which resulted in an amino acid change (glycine to valine), is expected to be probably harmful or destructive. An alteration in the protein structure is important for its interactions with other proteins or functionality [31].

*TLR9* is expressed in the endosomal compartment by neutrophils, plasmacytoid dendritic cells, and B and T lymphocytes [116]. There have been several studies establishing a significant association between the *TLR9* gene and PD in humans [117,118,119]. Thus, the possible association between genetic variations of *TLR9* and predisposition to canine PD has also become an important research topic. In this case, *rs375556098* and *rs201959275* polymorphisms in the *TLR9* gene may be interesting candidates as biomarkers of predisposition to the development of PD [32], and therefore reinforces the need for further research between these elements. The *rs22882109*, *rs852734185*, *rs851587151*, and *rs851944523* polymorphisms lead to a missense mutation, which changes a serine to a glycine, a leucine to a proline, a cysteine to an arginine, and a tryptophan to an arginine, respectively. These polymorphisms originate a non-conservative amino acid change, which can provoke a major effect [32].

Finally, the *RANK* gene is involved in the osteoclastogenesis regulation mechanism existing in PD [120] and is therefore an interesting element for the study of PD. In the case–control study reviewed here, it was not possible to find an association between polymorphic variation in the *RANK* gene and susceptibility to PD, which indicates that it may be important to study other polymorphic variations [33].

Given the complexity and the causal pathways of a multifactorial disease such as PD, the role of genetics is significant in its onset, progression, and severity [115,121]. All of this is because hundreds or thousands of genes can be associated with the progression of PD. In other words, this disease is not the result of a single mutation or a mutation in a single gene [122]. This conclusion can also be easily proven with the obtained results in this systematic review. In only two of the six genetic variations analyzed, possible biomarkers of susceptibility to the development of PD were identified.

Through the knowledge that exists to date, it is possible to understand that genetic variance between individuals is a factor to be considered in terms of susceptibility in various illnesses. PD is one of many examples, however, there are many others including those in veterinary medicine. Canine atopic dermatitis is a frequent allergic inflammatory skin condition that is very identical to human atopic dermatitis [123]. This pathology has been the target of studies, namely genome-wide association studies, in which it has been possible to identify single nucleotide polymorphisms associated with the susceptibility to develop canine atopic dermatitis [124,125]. However, there are other veterinary medical specialties with studies already conducted regarding the importance of genetic factors in certain diseases. In endocrinology, more specifically in diabetes mellitus, it is possible to verify that canine cytokine genes, those that control the immune balance TH1/TH2, may contribute to the vulnerability of some breeds in relation to this pathology [126]. Other research has concluded that certain single nucleotide polymorphisms in the *CTLA4* gene are associated with the development of canine diabetes mellitus in specific breeds [127]. In canine inflammatory bowel disease, one of the most challenging pathologies in gastroenterology, polymorphisms in the toll-like receptors 5 [128] and 4 [129] genes are believed to have a significant contribution as well as the major histocompatibility complex class II locus (dog leukocyte antigen) [130], specifically in German shepherds. Increasing research in this area has led to promising results, as described in the previous examples. In addition, diseases such as hip joint laxity [131], mammary tumor [132], mitral valve myxomatous disease [133], and primary cataracts in dogs [134] have also been the subject of study, which reinforce the importance of genetic profiling.

The genetic component is included in the etiology of many diseases in human medicine, and its impact varies greatly depending on the pathology. Thus, the impact that the genetic component has in terms of etiology is very important for the diagnosis and treatment of a certain disease, especially an early treatment recommended by a preventive medicine strategy. For example, it appears that the *IL1* composite genotype has an equivocal ability in detecting susceptibility to PD. Moreover, genetic tests are now available, namely the GenoType PST test (Hain Diagnostica, Germany; Greenstein and Hart 2002), which allows for the analysis of *IL1* gene polymorphisms and thus correlates with PD susceptibility [135]. However, its utility may be limited to only very specific populations at best [72]. Through these data, a parallel can be drawn with veterinary medicine and canine PD. It is known that there are dog breeds that are more susceptible to developing PD such as toy breeds, brachycephalic dogs, Maltese terriers, Schnauzers, and some variants of Greyhounds [7,12,136]. Thus, the variations that have been identified in the studies can have variable effects in different breeds [12]. This fact makes it increasingly important to direct future studies to specific breeds. Only in this way will it be possible to achieve a new approach, in which we will have clinical genomics at the service of preventive medicine. This will allow the implementation of medical and home care early and, consequently, will avoid the advanced stages of PD, especially in the most susceptible breeds. Therefore, the identification of genetic risk factors associated with PD is essential to improving the prevention and treatment strategies for this disease [137]. With this, it is expected that periodontal gene therapy will become part of everyday life, not only for physicians, but also for veterinarian practitioners.

In addition to all this, there is another very important factor that should be considered when promoting the development of studies on PD. Recent research in human medicine shows a clear association between chronic inflammatory states such as PD and the development of cancer [138,139,140,141]. In the future, it would be interesting to examine this relationship in veterinary medicine, based on canine PD.

## 5. Limitations

The authors of this review considered it is important to describe the limitations they encountered when writing this systematic review to ensure proper interpretation. The most significant limitation was the small number of studies on the topic. Moreover, all of these studies were case–control studies with a small number of cases and controls. Furthermore, the animals used in the groups were not of a defined breed. It will be important in the future to develop studies that analyze the relationship between specific breeds and the genetic variations found. Each element considered relevant to the knowledge of genomic medicine in canine PD had only one study regarding it. This fact makes it difficult to obtain conclusions with strong statistical power.

The search being conducted in only two databases may also be considered as a limitation, as studies of interest for this systematic review may have been lost. Nevertheless, if more databases had been used, it could have increased the unnecessary duplication of studies.

However, we believe that this study has made a significant contribution to the analysis of the relationship between genomic medicine and canine PD. Thus, the present review provided an overview of the current state-of-the-art and highlights the need for further research in this field of knowledge.

## 6. Conclusions

The present systematic review allows us to recognize potential biomarkers that may enable the early identification of individuals in a population that have genetic variations, namely simple nucleotide polymorphisms, which may make them more susceptible to the development of periodontal disease. Thus, two variations were found in the *TLR9* gene with this potential and eight variations in the *IL1* gene, seven of them in the *IL1A* gene and one in the *IL1B* gene.

Similar to what already occurs in human medicine, this scientific information allows for the development of diagnostic kits that, together with microbiome tests, could revolutionize the prophylaxis and treatment of the disease in the case of companion animals, namely in the case of dogs.

The obtained results in this review reinforce the importance of the genomic factors in the study of PD, not only in veterinary medicine, but also in human medicine, given the similarities between the two clinical conditions. It is easily understandable that a better characterization of the canine animal model, as an essential element in the study of this disease in humans, will contribute to the translational application to human medicine of all the knowledge obtained using it. From the reviewed studies, it is possible to see that there is only one team working on the subject of clinical genomics in canine PD, which reinforces the need for cooperation between several scientific areas.

Due to the scarcity of studies found in veterinary medicine and the advancements in human medicine, we believe that knowledge about genomic medicine will continue to expand. In the specific case of canine PD, it will allow for the promotion of oral health through an individualized strategy and in a preventive approach.

In terms of future perspectives, it is important to conduct further research on the already detected genetic polymorphisms through studies involving a larger number of individuals, which will enhance statistical robustness. Furthermore, since there are thousands of genes implicated in the progression of PD, many studies still need to be developed.

## Figures and Tables

**Figure 1 animals-13-02463-f001:**
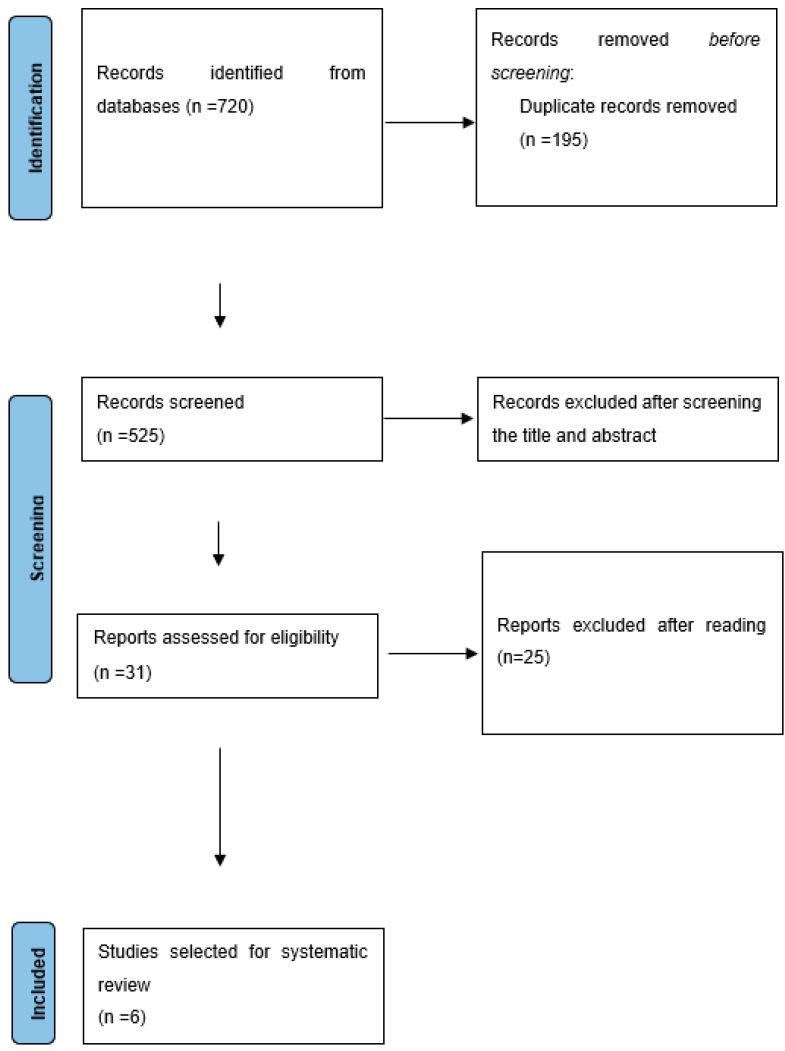
Flowchart of the identification, screening, and inclusion of studies in the systematic review based on the PRISMA 2020 guidelines.

**Table 1 animals-13-02463-t001:** Summary of the main characteristics of the reviewed articles (*n* = 6).

Study	Study Design	Study Sample	Study Variable	Objectives	Methodology	Main Results	Genetic Variations Detected	Alteration in the Conformation of Proteins
Morinha, F. et al., 2011 [12]	Case–control	70 dogs (45 controls and 25 cases)	*IL6* gene	Identify genetic variations of *IL6* gene and verify its association with PD. Clinical periodontal evaluation of the individuals.	General and oral (odonto-stomatological) examination. Collection of blood samples for subsequent DNA extraction. Amplification and genotyping of 2 regions of the dog *IL6* gene (5′UTR-exon 2 and exon 5–3′UTR), using the respective forward and reverse primers. The amplified fragments were purified and sequenced.	In the analyzed population, there was no evidence that the identified variants contributed significantly to PD susceptibility.	Three new single nucleotide polymorphisms: *I/2_g.564T>C*—exon 2 *I/5_g.105G>A*—exon 5 *I/5_g.440G>A*—exon 5	The sequence variant *I/5_g.105 G>A* led to an amino acid change (arginine to glutamine).
Morinha, F. et al., 2012 [29]	Case–control	90 dogs (50 controls and 40 cases)	*LTF* gene	Identify genetic variations of dog *LTF* gene and verify its association with PD.	General and oral (odonto-stomatological) examination. Collection of blood samples for subsequent DNA extraction. The regions of *LTF* analyzed were exon 12 and 15. For the polymerase chain reaction (PCR) amplification of these regions were selected specific primers. All amplified fragments were purified and sequenced.	Does not provide evidence that *LTF* variants contributed to the genetic basis of canine PD.	Eight new single nucleotide polymorphisms were detected: *L/2_g.288T>G*—intron 2 *L/2_g.403G>A*—intron 2 *L/15_g.213C>T*—intron 14 *L/15_g.411C>T* –exon 15 *L/15_g.420G>A*—exon 15 *L/15_g.445A>G*—exon 15 *L/15_g.482G>A*—exon 15 *L/15_g.514A>G*—exon 15	The sequence variant *L/15_g.411C>T* led to an amino acid change (proline to leucine).
Albuquerque, C. et al., 2014 [30]	Case–control	90 dogs (50 controls and 40 cases)	*IL10* gene	Identify *IL10* gene variations that can influence *IL10* regulatory role, and consequently the canine PD susceptibility.	General and oral (odonto-stomatological) examination. Collection of blood samples for subsequent DNA extraction. Analysis of two fragments, one corresponding to the target region in the 5′ flanking sequence and the other including 5′ untranslated region and exon 1, were obtained using forward and reverse primers. PCR amplifications and subsequent, the fragments were purified and analyzed.	No significant association was found between the *IL10* genetic variations and canine PD.	Six new nucleotide variations and a 3-nucleotide deletion: *IL10/1_g.276_278delTGA*–5′ flanking region *IL10/1_g.506A>g*—5′ flanking region *IL10/2_g.285G>A*—exon 1 (signal peptide) *IL10/2_g.305C>T*—exon 1 *IL10/2_g.424C>A*—intron 1 *IL10/2_g.497C>T*—intron 1 *IL10/2_g.513C>A*—intron 1	The variation *IL10/2_g.285G>A* led to an amino acid change (glycine to arginine) in the putative signal peptide.
Albuquerque, C. et al., 2015 [31]	Case–control	90 dogs (50 controls and 40 cases)	IL1-α (*IL1A*) and IL1-β (*IL1B*) genes	Identify genetic variations from the *IL1A* and *IL1B* genes and evaluate their possible association with PD.	General and oral (odonto-stomatological) examination. Collection of blood samples for subsequent DNA extraction. Two fragments of the *IL1A* gene were analyzed. Fragment 1 includes exon 2 and partial intron 1 and fragment 2 includes exon 5. Only 1 fragment was analyzed in the *IL1B* gene, which includes exons 4 and 5. All amplified fragments were purified and sequenced.	Genetic variations in *IL1* may play a significant role in canine PD susceptibility. *IL1A/1_g.388C* allele was associated with a decreased PD risk. *IL1A/1_g.521A* allele could confer an increased risk.	Eight single nucleotide polymorphisms identified: *IL1A/1_g.110A>G*—intron 1 *IL1A/1_g.113C>A*—intron 1 *IL1A/1_g.129G>A*—intron 1 *IL1A/1_g.388A>C*—intron 1 *IL1A/1_g.521T>A*—intron 1 *IL1A/1/2_g.153T>A*—intron 4 *IL1A/2_g.515G>T*—exon 5 *IL1B_g.525G>A*—exon 5	The genetic variation *IL1A/2_g.515G>T* resulted in a change of amino acid, glycine to valine.
Gonçalves-Anjo, N. et al., 2019 [32]	Case–control	90 dogs (50 controls and 40 cases)	*TLR 9* gene	Investigating the role that seven potential polymorphic sites from exon 3 of the gene *TLR9* play in predisposing to PD.	General and oral (odonto-stomatological) examination. Collection of blood samples for subsequent DNA extraction. A fragment of exon 3 was obtained with a forward and reverse primer and DNA was amplified. The amplified fragments were purified and subsequently sequenced.	Two of the genetic variants examined (*rs375556098* and *rs201959275*) may serve as biomarkers for the susceptibility of dogs to develop PD.	The selected region of exon 3 has seven hypothetical polymorphic sites: *rs851706751* *rs22882109* *rs852868639* *rs852734185* *rs851587151* *rs851944523* *rs22882111*	The *rs22882109, rs852734185, rs851587151* and *rs851944523* originated a non-conservative amino acid change.
Gonçalves-Anjo, N. et al., 2022 [33]	Case–control	90 dogs (50 controls and 40 cases)	*RANK* gene	Identification of genetic variations and potential associations between the *RANK* gene and PD.	General and oral (odonto-stomatological) examination. Collection of blood samples for subsequent DNA extraction. After that, the DNA was amplified by PCR, more specifically the fragment corresponding to exon 7. Finally, the DNA was sequenced and sequence variants were identified.	None of the four variations discovered were thought to have a significant impact on the clinical disease’s severity or in the possibility that the dogs would acquire PD.	Four new genetic variations in the intronic region of the fragment: *g.85A>G* *g.151G>T* *g.268A>G* *g.492T>C*	Genetic variations in the intronic region could not change the amino acid in the final protein.

**Table 2 animals-13-02463-t002:** Evaluation of the quality of the case–control studies that were included using the scale proposed by Nibali and collaborators (2013) [34].

Author	Selection (4 Items)	Comparability (1 Item)	Exposure (3 Items)	Study Design (4 Items)	Genetic Analysis (8 Items)
Morinha, F. et al., 2011 [12]	4/4	1/1	2/3	3/4	6/8
Morinha, F. et al., 2012 [29]	4/4	1/1	2/3	3/4	6/8
Albuquerque, C. et al., 2014 [30]	4/4	1/1	2/3	3/4	6/8
Albuquerque, C. et al., 2015 [31]	4/4	1/1	2/3	3/4	6/8
Gonçalves-Anjo, N. et al., 2019 [32]	4/4	1/1	2/3	3/4	6/8
Gonçalves-Anjo, N. et al., 2022 [33]	4/4	1/1	2/3	3/4	6/8

## Data Availability

Not applicable.

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
