# Peer review of "Genomic Medicine in Canine Periodontal Disease: A Systematic Review"

_animals, 2023, doi:10.3390/ani13152463_

Round 1

Reviewer 1 Report

Dear Authors

The summary part incorrectly states that periodontal disease is one of the most prevalent diseases in both companion animals and humans. While it is prevalent in both, it is not accurate to classify it as one of the most prevalent diseases.

The introduction provides a detailed and accurate description of periodontal disease, the factors that contribute to the disease, and the understanding of genetics in disease development. All the information in this paragraph is based on current scientific research and appropriately referenced.

The methodology is clearly described, including the inclusion and exclusion criteria, information sources, search strategy, study selection and data extraction process, as well as the assessment of bias in individual studies. This is a standard approach for conducting systematic reviews and meta-analyses.

The results are presented meticulously and include:

01.Search and exclusion process: The number of studies found, reasons for exclusion, and the final number of studies included in the systematic review are described in detail.

02. Assessment of studies: The authors provide information on how they evaluated and analyzed the studies, including identifying the important factors to assess and classifying the studies based on their design.

03. Detailed findings from each study: The authors provide detailed information on each study included in the systematic review. This includes information about the study sample, the variables studied, the main results, and the detected gene variants.

However, the presentation of the studies on canine periodontal disease is limited, citing only six studies. Detailed information about these studies, such as their objectives, methodology, and results, is not fully presented.

Discussion: The paragraphs lacked clear organization and coherence. There should be better structuring of ideas and logical flow between sentences to enhance readability and comprehension. The use of technical terms and complex language without sufficient explanation may make it challenging for general readers to understand the content.

The paragraph provides a concise but not detailed description of periodontal disease (PD) and its impact on overall health. Some specific conditions related to the liver, lungs, heart and kidneys, bone inflammation, and malnutrition in infants are mentioned but not clearly presented in terms of the specific impact of PD on each condition.

The discussion on the role of genetic factors in PD in humans is concise and does not provide specific information on the scientific studies that have demonstrated a strong association between genetic factors and PD in humans.

Some information on scientific research methods such as "omics" analysis and the use of animal models in PD research is presented in a general manner without going into detail about the methods and specific benefits of these approaches in understanding PD.

The presentation of the studies conducted on dogs regarding PD is limited, citing only six studies. Detailed information about these studies, including their objectives, methods, and results, is not fully presented.

The information about genetic factors and their role in PD development and severity is presented clearly. Relevant studies on genetics and PD in humans and dogs are mentioned, and their findings are accurately presented.

One potential drawback is the citation of some human genetics studies without providing specific information about their results. Additionally, the application of results from human genetics studies to studying PD in dogs is mentioned in a general manner without delving into the methods and findings of these studies.

The conclusion should be revised to provide a concise summary of the main findings and contributions of the systematic review. 

Best wishes

Here are some grammar errors.

Sentences 94: "The aim of Genomic Medicine is to understand genetic factors in order to establish a higher or lower predisposition to the development of certain."

-> Edit to: "The aim of Genomic Medicine is to understand genetic factors that establish a higher or lower predisposition to the development of certain diseases."

Sentences 95: "Only in this way it will be possible to develop an individualized and more accurate clinical approach."

-> Edit to: "Only in this way, will it be possible to develop an individualized and more accurate clinical approach."

Sentences 108: "Given the existence of few studies about genomics medicine in canine PD, this review will certainly contribute to the knowledge of which genetic factors have already been analyzed and, consequently, promote further research on other potential biomarkers."

-> Edit to:"Given the limited number of studies about genomic medicine in canine PD, this review will certainly contribute to the understanding of which genetic factors have already been analyzed and, consequently, promote further research on other potential biomarkers."

Sentences (113-115): "The PRISMA 2020 (Preferred Reporting Items for Systematic Reviews and Meta-Analyses) guidelines [30] was used to conduct the writing of this paper."

-> Edit to: "The PRISMA 2020 (Preferred Reporting Items for Systematic Reviews and Meta-Analyses) guidelines [30] were used to conduct the writing of this paper."

Sentences (125-126): "Considering these criteria, the selected studies were read in full and, after reading, only those which contained explicit information were considered."

-> Edit to: "Considering these criteria, we read the selected studies in full and only considered those that contained explicit information."

Sentences (127-130): "The studies were searched in the electronic databases Science Direct/Elsevier and PubMed (National Library of Medicine, National Institutes of Health, Bethesda, MD, USA), between 1 January 2002 and 31 December 2022 ."

-> Edit to: "We searched for studies in the electronic databases Science Direct/Elsevier and PubMed (National Library of Medicine, National Institutes of Health, Bethesda, MD, USA), between 1 January 2002 and 31 December 2022."

Sentences 152: "Only by evaluating the title and respectively abstract, it was possible to exclude 494 studies." - This sentence is slightly cumbersome due to the use of the passive voice. A better articulation could be "By evaluating only the title and respective abstracts, we were able to exclude 494 studies."

Sentence 156: "After reading the 31 studies, only six studies remains which met all the criteria defined from the beginning." - This sentence requires a structural adjustment. "After reviewing the 31 studies, only six that still met all the initially defined criteria" would be more natural.

Sentence 161: "The aspects considered essential in the evaluation of these studies were the development of scientific articles with an exclusive focus on the genomics medicine of canine PD." - This sentence is somewhat cumbersome and could be sentence. "We consider the development of scientific articles focusing exclusively on canine PD genomics medicine as an essential aspect in the evaluation of these studies." would be clearer and understand.

Sentence 165: "Additionally, this review demonstrates some heterogeneity between studies, either by the element under analysis or by the final conclusion." - This sentence could be made clearer by using a precise sentence structure. "Additionally, this review demonstrates some heterogeneity between studies, as evidenced by the elements under analysis or the final conclusions." would be better.

Sentence 181: "Two regions of the IL6 gene were amplified by PCR, and all of the amplified fragments were sequence and genotyped,...". The word "sequence" should be replaced by "sequenced".

Sentence 198: "...and no association between the genetic variation and the susceptibility to PD was established.". The word "genetic variation" should be replaced by "genetic variations".

Sentence 209: "No significant differences were detected in genotype and allele frequencies of the seven new genetic variations between the PD cases and control groups." The article "the" should be added before "genotype" and "allele frequencies".

Sentence 238: "However, this study allowed the authors to discover four new intronic variations that were banked in GenBank." The word "banked" should be replaced by "stored" or "deposited".

Sentence 241: "Table 2 shows the results of the risk of bias analysis for each of the six selected case-control studies...". The article "the" should be added before "risk of bias analysis".

Sentence 317: "IL6 is a pleiotropic cytokine synthesized in diverse cells, like neutrophils and macrophages, and has functions in the acute phase protein production, among others" should be revised to "IL6 is a pleiotropic cytokine synthesized in various cells, such as neutrophils and macrophages, and it plays a role in the production of acute phase proteins, among other functions."

Sentence 339: "Nevertheless, one of the genetic variations located in exon 15, L/15_g.411C>T, led to the substitution of proline to leucine, which has an amino acid change that would be expected to be potentially harmful to the LTF protein" should be corrected to "Nevertheless, one of the genetic variations located in exon 15, L/15_g.411C>T, led to the substitution of proline with leucine, resulting in an amino acid change that could potentially affect the LTF protein."

Sentence 317: "IL6 is a pleiotropic cytokine synthesized in diverse cells, like neutrophils and macrophages, and has functions in the acute phase protein production, among others" should be revised to "IL6 is a pleiotropic cytokine synthesized in various cells, such as neutrophils and macrophages, and it plays a role in the production of acute phase proteins, among other functions."

Sentence 339: "Nevertheless, one of the genetic variations located in exon 15, L/15_g.411C>T, led to the substitution of proline to leucine, which has an amino acid change thatwould be expected to be potentially harmful to the LTF protein " should be corrected to "Nevertheless, one of the genetic variations located in exon 15, L/15_g.411C>T, led to the substitution of proline with leucine, resulting in an amino acid change that could potentially affect the LTF protein."

Sentence 389: "All of this because hundreds or thousands of genes can be associated with the progression of PD, that is, this disease is not the result of a single mutation or mutation in a single gene" should be revised to "All of this is because hundreds or thousands of genes can be associated with the progression of PD. In other words, this disease is not the result of a single mutation or a mutation in a single gene."

Sentences 453: "The search in only two databases may also be considered a limitation" - This paragraph needs to be edited to ensure grammatical accuracy, for example: "The search conducted in only two databases may also be considered a limitation. "

Sentences 455: "Nevertheless, we consider that if more databases had been used this would lead to an unjustified number of duplicates." - It is necessary to revise the sentence structure to avoid ambiguity, for example: "However, if more databases had been used, it could have caused in an increased number of duplicate studies, which would not be justified."

Sentences 457: "Thus, it was possible to review the state of the art and demonstrate the need for future research in this same area." - This paragraph could be improved to avoid repetition, for example: "Thus, this review has provided an overview of the current state of the art and highlighted the need for further research in the field."

Sentences 470: "Due to the scarce studies found in veterinary medicine and the existing advances in human medicine, we believe that knowledge about genomic medicine will increase more and more." - Need to edit the sentence to be clearer, for example: "Considering the limited number of studies in veterinary medicine and the significant advancements in human medicine, it is anticipated that knowledge about genomic medicine will continue to expand."

Sentence 444: "The authors of this review consider it important to describe the limitations they 445 encountered when writing this systematic review, as only then will it be possible to interpret it properly." This sentence does not reflect reality. Edit to: "The authors of this review consider it important to describe the limitations they encountered when writing this systematic review to ensure proper interpretation."

Sentence 448: "Moreover, all of these studies were case-control studies and have a small number 448 of cases and controls." - Sentence structure is incorrect. Edit to: "Moreover, all of these studies were case-control studies with a small number of cases and controls."

Sentence 455: "Nevertheless, we consider that if 455 more databases had been used this would lead to an unjustified number of duplicates." - Sentence structure is incorrect. Change it to: "Nevertheless, if more databases had been used, it could have been increased in unnecessary duplication of studies."

Sentence 461: "The obtained results in this review reinforce the importance of the genomic compo- 462 nent in the study of PD, not only in veterinary medicine but also in human medicine, given 463 the similarities between the two clinical conditions." - The term "genomic component" is ambiguous and needs to be defined more clearly. Edit to: "The obtained results in this review reinforce the importance of genetic factors in the study of PD, not only in veterinary medicine but also in human medicine, given the similarities between the two clinical conditions."

Sentence 469: "Due to the scarce studies found in veterinary medicine and the existing advances in 470 human medicine, we believe that knowledge about genomic medicine will increase more 471 and more." - Sentence structure is incorrect. Edit to: "Due to the scarcity of studies found in veterinary medicine and the advancements in human medicine, we believe that knowledge about genomic medicine will continue to expand."

Sentence 474: "In terms of future work perspectives, it is important that there is more research on 474 the genetic polymorphisms already detected, through studies with a larger number of in- 475 dividuals, which leads to greater statistical robustness." - Sentence structure is incorrect. Edit to: "In terms of future perspectives, it is important to conduct further research on the already detected genetic polymorphisms through studies involving a larger number of individuals, which will enhance statistical robustness."

Best wishes

Author Response

Response to Reviewer #1

Comments and Suggestions for Authors

Dear Authors

The summary part incorrectly states that periodontal disease is one of the most prevalent diseases in both companion animals and humans. While it is prevalent in both, it is not accurate to classify it as one of the most prevalent diseases.

Authors: Dear reviewer 1, thank you for your note. The sentence has been changed accordingly.

The introduction provides a detailed and accurate description of periodontal disease, the factors that contribute to the disease, and the understanding of genetics in disease development. All the information in this paragraph is based on current scientific research and appropriately referenced.

The methodology is clearly described, including the inclusion and exclusion criteria, information sources, search strategy, study selection and data extraction process, as well as the assessment of bias in individual studies. This is a standard approach for conducting systematic reviews and meta-analyses

The results are presented meticulously and include:

01.Search and exclusion process: The number of studies found, reasons for exclusion, and the final number of studies included in the systematic review are described in detail.

  1. Assessment of studies: The authors provide information on how they evaluated and analyzed the studies, including identifying the important factors to assess and classifying the studies based on their design.
  2. Detailed findings from each study: The authors provide detailed information on each study included in the systematic review. This includes information about the study sample, the variables studied, the main results, and the detected gene variants.

Authors:The authors are recognized for the opinion that the reviewer expresses about the work.

However, the presentation of the studies on canine periodontal disease is limited, citing only six studies. Detailed information about these studies, such as their objectives, methodology, and results, is not fully presented. 

Authors: The authors agree with the reviewer. In Table 1 we have added the objective and methodology of each of the selected articles, since the main results were already described.

Discussion: The paragraphs lacked clear organization and coherence. There should be better structuring of ideas and logical flow between sentences to enhance readability and comprehension. The use of technical terms and complex language without sufficient explanation may make it challenging for general readers to understand the content.

Authors: We hope that the current version corresponds to the reviewer's aspirations, to whom we thank the enlightened contribution.

The paragraph provides a concise but not detailed description of periodontal disease (PD) and its impact on overall health. Some specific conditions related to the liver, lungs, heart and kidneys, bone inflammation, and malnutrition in infants are mentioned but not clearly presented in terms of the specific impact of PD on each condition. 

Authors: The authors thank you for your comment. In order to address this topic we have detailed the description of periodontal disease in the introduction. Regarding the systemic implications, these have been detailed in the discussion.

The discussion on the role of genetic factors in PD in humans is concise and does not provide specific information on the scientific studies that have demonstrated a strong association between genetic factors and PD in humans.

Authors: The authors agree with the reviewer. Several studies demonstrating this same association were referred to in line 285.

Some information on scientific research methods such as "omics" analysis and the use of animal models in PD research is presented in a general manner without going into detail about the methods and specific benefits of these approaches in understanding PD.

Authors: In fact, the information regarding the multi-omics analysis was briefly addressed in the first version of the manuscript. At the moment, the authors have added more information that they consider relevant. 

The presentation of the studies conducted on dogs regarding PD is limited, citing only six studies. Detailed information about these studies, including their objectives, methods, and results, is not fully presented. 

Authors: We hope that the current version corresponds to the reviewer's aspirations, to whom we thank the enlightened contribution.

The information about genetic factors and their role in PD development and severity is presented clearly. Relevant studies on genetics and PD in humans and dogs are mentioned, and their findings are accurately presented.

One potential drawback is the citation of some human genetics studies without providing specific information about their results. Additionally, the application of results from human genetics studies to studying PD in dogs is mentioned in a general manner without delving into the methods and findings of these studies.  

Authors: We hope that the current version corresponds to the reviewer's aspirations, to whom we thank the enlightened contribution.

The conclusion should be revised to provide a concise summary of the main findings and contributions of the systematic review.

Authors: We hope that the current version corresponds to the reviewer's aspirations, to whom we thank the enlightened contribution.

Comments on the Quality of English Language

Here are some grammar errors.

Sentences 94: "The aim of Genomic Medicine is to understand genetic factors in order to establish a higher or lower predisposition to the development of certain."

-> Edit to: "The aim of Genomic Medicine is to understand genetic factors that establish a higher or lower predisposition to the development of certain diseases.

Sentences 95: "Only in this way it will be possible to develop an individualized and more accurate clinical approach."

-> Edit to: "Only in this way, will it be possible to develop an individualized and more accurate clinical approach."

Sentences 108: "Given the existence of few studies about genomics medicine in canine PD, this review will certainly contribute to the knowledge of which genetic factors have already been analyzed and, consequently, promote further research on other potential biomarkers."

-> Edit to:"Given the limited number of studies about genomic medicine in canine PD, this review will certainly contribute to the understanding of which genetic factors have already been analyzed and, consequently, promote further research on other potential biomarkers."

Sentences (113-115): "The PRISMA 2020 (Preferred Reporting Items for Systematic Reviews and Meta-Analyses) guidelines [30] was used to conduct the writing of this paper."

-> Edit to: "The PRISMA 2020 (Preferred Reporting Items for Systematic Reviews and Meta-Analyses) guidelines [30] were used to conduct the writing of this paper."

Sentences (125-126): "Considering these criteria, the selected studies were read in full and, after reading, only those which contained explicit information were considered."

-> Edit to: "Considering these criteria, we read the selected studies in full and only considered those that contained explicit information."

Sentences (127-130): "The studies were searched in the electronic databases Science Direct/Elsevier and PubMed (National Library of Medicine, National Institutes of Health, Bethesda, MD, USA), between 1 January 2002 and 31 December 2022 ."

-> Edit to: "We searched for studies in the electronic databases Science Direct/Elsevier and PubMed (National Library of Medicine, National Institutes of Health, Bethesda, MD, USA), between 1 January 2002 and 31 December 2022."

Sentences 152: "Only by evaluating the title and respectively abstract, it was possible to exclude 494 studies." - This sentence is slightly cumbersome due to the use of the passive voice. A better articulation could be "By evaluating only the title and respective abstracts, we were able to exclude 494 studies."

Sentence 156: "After reading the 31 studies, only six studies remains which met all the criteria defined from the beginning." - This sentence requires a structural adjustment. "After reviewing the 31 studies, only six that still met all the initially defined criteria" would be more natural.

Sentence 161: "The aspects considered essential in the evaluation of these studies were the development of scientific articles with an exclusive focus on the genomics medicine of canine PD." - This sentence is somewhat cumbersome and could be sentence. "We consider the development of scientific articles focusing exclusively on canine PD genomics medicine as an essential aspect in the evaluation of these studies." would be clearer and understand.

Sentence 165: "Additionally, this review demonstrates some heterogeneity between studies, either by the element under analysis or by the final conclusion." - This sentence could be made clearer by using a precise sentence structure. "Additionally, this review demonstrates some heterogeneity between studies, as evidenced by the elements under analysis or the final conclusions." would be better.

Sentence 181: "Two regions of the IL6 gene were amplified by PCR, and all of the amplified fragments were sequence and genotyped,...". The word "sequence" should be replaced by "sequenced".

Sentence 198: "...and no association between the genetic variation and the susceptibility to PD was established.". The word "genetic variation" should be replaced by "genetic variations".

Sentence 209: "No significant differences were detected in genotype and allele frequencies of the seven new genetic variations between the PD cases and control groups." The article "the" should be added before "genotype" and "allele frequencies"

Sentence 238: "However, this study allowed the authors to discover four new intronic variations that were banked in GenBank." The word "banked" should be replaced by "stored" or "deposited".

Sentence 241: "Table 2 shows the results of the risk of bias analysis for each of the six selected case-control studies...". The article "the" should be added before "risk of bias analysis".

Sentence 317: "IL6 is a pleiotropic cytokine synthesized in diverse cells, like neutrophils and macrophages, and has functions in the acute phase protein production, among others" should be revised to "IL6 is a pleiotropic cytokine synthesized in various cells, such as neutrophils and macrophages, and it plays a role in the production of acute phase proteins, among other functions."

Sentence 339: "Nevertheless, one of the genetic variations located in exon 15, L/15_g.411C>T, led to the substitution of proline to leucine, which has an amino acid change that would be expected to be potentially harmful to the LTF protein" should be corrected to "Nevertheless, one of the genetic variations located in exon 15, L/15_g.411C>T, led to the substitution of proline with leucine, resulting in an amino acid change that could potentially affect the LTF protein."

Sentence 317: "IL6 is a pleiotropic cytokine synthesized in diverse cells, like neutrophils and macrophages, and has functions in the acute phase protein production, among others" should be revised to "IL6 is a pleiotropic cytokine synthesized in various cells, such as neutrophils and macrophages, and it plays a role in the production of acute phase proteins, among other functions."

Sentence 339: "Nevertheless, one of the genetic variations located in exon 15, L/15_g.411C>T, led to the substitution of proline to leucine, which has an amino acid change thatwould be expected to be potentially harmful to the LTF protein " should be corrected to "Nevertheless, one of the genetic variations located in exon 15, L/15_g.411C>T, led to the substitution of proline with leucine, resulting in an amino acid change that could potentially affect the LTF protein."

Sentence 389: "All of this because hundreds or thousands of genes can be associated with the progression of PD, that is, this disease is not the result of a single mutation or mutation in a single gene" should be revised to "All of this is because hundreds or thousands of genes can be associated with the progression of PD. In other words, this disease is not the result of a single mutation or a mutation in a single gene."

Sentences 453: "The search in only two databases may also be considered a limitation" - This paragraph needs to be edited to ensure grammatical accuracy, for example: "The search conducted in only two databases may also be considered a limitation. "

Sentences 455: "Nevertheless, we consider that if more databases had been used this would lead to an unjustified number of duplicates." - It is necessary to revise the sentence structure to avoid ambiguity, for example: "However, if more databases had been used, it could have caused in an increased number of duplicate studies, which would not be justified."

Sentences 457: "Thus, it was possible to review the state of the art and demonstrate the need for future research in this same area." - This paragraph could be improved to avoid repetition, for example: "Thus, this review has provided an overview of the current state of the art and highlighted the need for further research in the field."

Sentences 470: "Due to the scarce studies found in veterinary medicine and the existing advances in human medicine, we believe that knowledge about genomic medicine will increase more and more." - Need to edit the sentence to be clearer, for example: "Considering the limited number of studies in veterinary medicine and the significant advancements in human medicine, it is anticipated that knowledge about genomic medicine will continue to expand."

Sentence 444: "The authors of this review consider it important to describe the limitations they 445 encountered when writing this systematic review, as only then will it be possible to interpret it properly." This sentence does not reflect reality. Edit to: "The authors of this review consider it important to describe the limitations they encountered when writing this systematic review to ensure proper interpretation."

Sentence 448: "Moreover, all of these studies were case-control studies and have a small number 448 of cases and controls." - Sentence structure is incorrect. Edit to: "Moreover, all of these studies were case-control studies with a small number of cases and controls."

Sentence 455: "Nevertheless, we consider that if 455 more databases had been used this would lead to an unjustified number of duplicates." - Sentence structure is incorrect. Change it to: "Nevertheless, if more databases had been used, it could have been increased in unnecessary duplication of studies.

Sentence 461: "The obtained results in this review reinforce the importance of the genomic compo- 462 nent in the study of PD, not only in veterinary medicine but also in human medicine, given 463 the similarities between the two clinical conditions." - The term "genomic component" is ambiguous and needs to be defined more clearly. Edit to: "The obtained results in this review reinforce the importance of genetic factors in the study of PD, not only in veterinary medicine but also in human medicine, given the similarities between the two clinical conditions."

Sentence 469: "Due to the scarce studies found in veterinary medicine and the existing advances in 470 human medicine, we believe that knowledge about genomic medicine will increase more 471 and more." - Sentence structure is incorrect. Edit to: "Due to the scarcity of studies found in veterinary medicine and the advancements in human medicine, we believe that knowledge about genomic medicine will continue to expand."

Sentence 474: "In terms of future work perspectives, it is important that there is more research on 474 the genetic polymorphisms already detected, through studies with a larger number of in- 475 dividuals, which leads to greater statistical robustness." - Sentence structure is incorrect. Edit to: "In terms of future perspectives, it is important to conduct further research on the already detected genetic polymorphisms through studies involving a larger number of individuals, which will enhance statistical robustness."

Authors: We thank the reviewer for his generosity in correcting the work in terms of the use of the English language. We proceed with all changes as recommended.

Reviewer 2 Report

Overall ,I think that this warrants publication as it shows the paucity of research on periodontal genomics that is out there and the limited research is well-described in results. Yet the introduction and discussion require major editing and conclusions need to be made from what was found from the systemic review. Below I also have editorial changes that I suggest throughout. 

Line 52: Recommended rewording to “ PD is plaque-induced inflammation of the periodontium which includes the gingiva, cementum, and periodontal ligament, and alveolar bone” … as it currently reads it is a little confusing. 

Line 54: technically they are not totally distinct as gingivitis is always present with periodontis is. So you could reword to say something like “PD is progressive, initially presenting as gingivitis and progressing

Line 65:  typo – should be serious, not seriously  

Line 70-96: I will leave this to the edior to make the final decision on the audience for this article but I do not think you need to explain the absics of genomics in a research article on genomics. I think that most already know the basics if htye are reading the article. 

Line 99 – you need a reference here, why is the dog a particular good model for clinical genomics?

Line 163 – remove the “by pure chance” 

Line 248 – developed should be changed to occurring

Line 250 – should change to may have serious health implications – the research linking systemic illness to PD in canines is vague at best  

Line 250-252 – this is an overly broad categorization on the effects of PD on systemic health which are inaccurate to our current knowldg base in dogs. I think if you truly want to touch on this you need to expand and discuss what is actually seen/proven in humans versus the few studies that exist in dogs suggesting that similar changes may occur. Otherwise this sentence should be removed

Line 258 – rather than saying different studies, you can say systematic reviews to clarify that is what you are referencing 

Line 259 – you do not need to define a cytokine 

Line 262 – you say for the various reasons already described – but you have not described anything yet. You can expand to describe the role of cytokines in PD briefly and then lead into why they are so impactful in PD genomics research 

Line 280-292: this paragraph is very repetitive and does not truly explain why the dog is a good model, rather than repeating that the dog is “ideal” you should expand on why.

Line 317-343: this is great! However, it is confusing as sometimes throughout the article you speak in a tone like this is for a novice (i.e. defining a cytokine) and then you switch to a scientific level explanation such as your description of the role of IL6 in PD and the importance it may play – I will leave this to the editor to make the final decision, but given the topic, I think the later is the more appropriate tone and I would recommend that you edit the manuscript so that it is all at this level. 

Line 344 – why is IL-10 “crucial”, expand. 

Overall I think this study is important and the systemic review was well done and the results are presented clearly. However, the introduction and discussion need substantial editing. More time needs to be spent discussing “why” this dog is an ideal model and what research we have (or do not have) that show parallels between the clinical course and more importantly the cytokine pathways of severe periodontal inflammation between the two species. I think more time should also be spent discussing the importance and role of cytokines and distinct cytokine alterations in PD progression and why this is likely impactful on why only certain genetic alterations rather than whole genome sequencing or RNA seq have been performed in the canine available studies. You also switch between your tone in the article – and I think that a scientific tone (not a GP level tone) should be maintained throughout the article. 

English is good, just some minor editing suggestions throughout

Author Response

Response to Reviewer #2

Comments and Suggestions for Authors

Overall,I think that this warrants publication as it shows the paucity of research on periodontal genomics that is out there and the limited research is well-described in results. Yet the introduction and discussion require major editing and conclusions need to be made from what was found from the systemic review. Below I also have editorial changes that I suggest throughout.

Line 52: Recommended rewording to “PD is plaque-induced inflammation of the periodontium which includes the gingiva, cementum, and periodontal ligament, and alveolar bone” … as it currently reads it is a little confusing.

Authors: In the presentation of periodontal disease, we took into account the valuable contribution of the reviewer 2, whom we thank.

Line 54: technically they are not totally distinct as gingivitis is always present with periodontis is. So you could reword to say something like “PD is progressive, initially presenting as gingivitis and progressing.

Authors: In the presentation of periodontal disease, we took into account the valuable contribution of the reviewer, whom we thank.

Line 65:  typo – should be serious, not seriously

Authors: The sentence has been changed accordingly.

Line 70-96: I will leave this to the edior to make the final decision on the audience for this article but I do not think you need to explain the absics of genomics in a research article on genomics. I think that most already know the basics if htye are reading the article.

Authors: The paragraph has been deleted as recommended.

Line 99 – you need a reference here, why is the dog a particular good model for clinical genomics?

Authors: A related reference has been added to the text.

Line 163 – remove the “by pure chance” 

Authors: The sentence has been changed accordingly.

Line 248 – developed should be changed to occurring

Authors: The sentence has been changed accordingly.

Line 250 – should change to may have serious health implications – the research linking systemic illness to PD in canines is vague at best  

Authors: The sentence has been changed accordingly.

Line 250-252 – this is an overly broad categorization on the effects of PD on systemic health which are inaccurate to our current knowldg base in dogs. I think if you truly want to touch on this you need to expand and discuss what is actually seen/proven in humans versus the few studies that exist in dogs suggesting that similar changes may occur. Otherwise this sentence should be remove 

Authors: To correspond to the request, the text of the discussion was enriched in this point.

Line 258 – rather than saying different studies, you can say systematic reviews to clarify that is what you are referencing 

Authors: The sentence has been changed accordingly.

Line 259 – you do not need to define a cytokine   

Authors: The sentence has been changed accordingly.

Line 262 – you say for the various reasons already described – but you have not described anything yet. You can expand to describe the role of cytokines in PD briefly and then lead into why they are so impactful in PD genomics research. 

Authors: To correspond to the request, the text of the discussion was enriched in this point.

Line 280-292: this paragraph is very repetitive and does not truly explain why the dog is a good model, rather than repeating that the dog is “ideal” you should expand on why. 

Authors: The requested explanation has been added to the discussion.

Line 317-343: this is great! However, it is confusing as sometimes throughout the article you speak in a tone like this is for a novice (i.e. defining a cytokine) and then you switch to a scientific level explanation such as your description of the role of IL6 in PD and the importance it may play – I will leave this to the editor to make the final decision, but given the topic, I think the later is the more appropriate tone and I would recommend that you edit the manuscript so that it is all at this level. 

Line 344 – why is IL-10 “crucial”, expand.  

Authors: To correspond to the request, the text of the discussion was enriched in this point.

Overall I think this study is important and the systemic review was well done and the results are presented clearly. However, the introduction and discussion need substantial editing. More time needs to be spent discussing “why” this dog is an ideal model and what research we have (or do not have) that show parallels between the clinical course and more importantly the cytokine pathways of severe periodontal inflammation between the two species. I think more time should also be spent discussing the importance and role of cytokines and distinct cytokine alterations in PD progression and why this is likely impactful on why only certain genetic alterations rather than whole genome sequencing or RNA seq have been performed in the canine available studies. You also switch between your tone in the article – and I think that a scientific tone (not a GP level tone) should be maintained throughout the article.

Authors: We hope that the current version corresponds to the reviewer's aspirations, to whom we thank the enlightened contribution.

Comments on the Quality of English Language

English is good, just some minor editing suggestions throughout.

Reviewer 3 Report

The paper is a review dealing with the genetic regulation of canine periodontal disease. It brings some new knowledge, but corrections are necessary.

Introduction: delete the text in rr. 74-93 (A gene is a DNA…..therapy methods). It is suitable for a textbook, not for paper in scientific magazine.

MM section: consider the relevance of chapter 2.3. It could be omitted without the loss of quality.

Discussion: main reservation. The text must be shortened at 50% at least. It is too wordy, repeats the information form Results and Tables.

Why did not you mention the Genome Wide Association Study (GWAS)? It is the method often used in such research. Were made GWAS for other diseases in dog? This information must be added.

Formal errors:

In Abstract, delete Background, Methods, Results, Conclusions.

Throughout the text, check the writing of numbers of references in normal type, not in superscript.

Throughout the text, check the writing of gene abbreviations in italic.

Author Response

Response to Reviewer #3

Comments and Suggestions for Authors

The paper is a review dealing with the genetic regulation of canine periodontal disease. It brings some new knowledge, but corrections are necessary.

Introduction: delete the text in rr. 74-93 (A gene is a DNA…..therapy methods). It is suitable for a textbook, not for paper in scientific magazine.

Authors: The paragraph has been deleted as recommended bw Reviewer 3.

MM section: consider the relevance of chapter 2.3. It could be omitted without the loss of quality. 

Authors: We delete this point as suggested.

Discussion: main reservation. The text must be shortened at 50% at least. It is too wordy, repeats the information form Results and Tables.

Authors: We hope to have amended the text in accordance with the reviewer's recommendations.

Why didn't you mention the Genome Wide Association Study (GWAS)? It is the method often used in such research. Were made GWAS for other diseases in dog? This information must be added.

Authors: We are grateful for the pertinent observation of the reviewer. We insert information related to this important point.

Formal errors:

In Abstract, delete Background, Methods, Results, Conclusions.

Throughout the text, check the writing of numbers of references in normal type, not in superscript.

Throughout the text, check the writing of gene abbreviations in italic. 

Authors: Formal errors were corrected as indicated by the reviewer.

Round 2

Reviewer 2 Report

Thank you for your many changes to the paper, here are some additional minor revisions I recommend 

Line 54 :It is a progressive disease

Line 75: briefly expand which characteristics make it ideal

Line 80: should genome wide association be abbreviated to GMA since it it used frequently in the paragraphs to follow?

Line 90-99: I think this can be condensed into one paragraph. GMA have been performed in different disease of the dog including canine systemic lupus erythematous related disease (23), heritable osteosarcoma (24), and squamous cell carcinoma of the digit (25). Further canine work has centered on the importance of using a reference panel with 365 genome sequences to improve the power of GMA studies (26) and performing meta-analysis after breed specific GWAS (27).

Line 100 – are you only focused on GMA in perio? If so – you should clarify

Line 146 – typo only six still met (delete the that)

Line 263 – I would add a line explaining that despite this a causative relationship has never been proven, and the relationship is largely explorative similar to your comment on renal disease.

Line 268 – do not need to clarify in humans again in relation to pre-term babies

Line 276 – I would wrap this section up by stating . The association of periodontal disease with systemic health is an impetus for identifying genetic variations that put patients at risk.

Line 277-297 ** this should all be moved down to below the omics blurb (line 341) then can go from this about the role of cytokines/human work into the animal work.

LINe 288 – new paragraph

Line 291 – you still need to expand more, which cytokines are most impactful in acute versus chornic periodontitis.

Line 529 – typo in characterization

minor grammatical/text errors

Author Response

Dear reviewer,

We appreciate all your valuable contributions that will allow us to have an improved version of our work. All comments and suggestions have been incorporated into the text and are underlined in green.

Line 54: It is a progressive disease - Solved

Line 75: briefly expand which characteristics make it ideal - This idea is clarified in the discussion.

Line 80: should genome wide association be abbreviated to GMA since it it used frequently in the paragraphs to follow? Done

Line 90-99: I think this can be condensed into one paragraph. GMA have been performed in different disease of the dog including canine systemic lupus erythematous related disease (23), heritable osteosarcoma (24), and squamous cell carcinoma of the digit (25). Further canine work has centered on the importance of using a reference panel with 365 genome sequences to improve the power of GMA studies (26) and performing meta-analysis after breed specific GWAS (27). We agree, thanks a lot.

Line 100: are you only focused on GMA in perio? If so – you should clarify

Line 146: typo only six still met (delete the that) - Solved

Line 263: I would add a line explaining that despite this a causative relationship has never been proven, and the relationship is largely explorative similar to your comment on renal disease.

Line 268: do not need to clarify in humans again in relation to pre-term babies - Done

Line 276: I would wrap this section up by stating . The association of periodontal disease with systemic health is an impetus for identifying genetic variations that put patients at risk. Done

Line 277-297: ** this should all be moved down to below the omics blurb (line 341) then can go from this about the role of cytokines/human work into the animal work. Done

LINe 288: new paragraph - done

Line 291: you still need to expand more, which cytokines are most impactful in acute versus chornic periodontitis. Done

Line 529:  typo in characterization - Solved

Reviewer 3 Report

The authors followed the reviewer´s comments. The Discussion section was bettered.

Formal comments: erase r. 128. Change the orientation of Table 1. Check r. 529 (characterization).

Author Response

Dear reviewer,

We appreciate all your valuable contributions that will allow us to have an improved version of our work. All comments and suggestions have been incorporated into the text and are underlined in green.
